# Bone Metabolism Effects of Medical Therapy in Advanced Renal Cell Carcinoma

**DOI:** 10.3390/cancers15020529

**Published:** 2023-01-15

**Authors:** Rosa Maria Paragliola, Francesco Torino, Agnese Barnabei, Giovanni Maria Iannantuono, Andrea Corsello, Pietro Locantore, Salvatore Maria Corsello

**Affiliations:** 1Department of Translational Medicine and Surgery, Unit of Endocrinology, Università Cattolica del Sacro Cuore—Fondazione Policlinico “Gemelli” IRCCS, Largo Gemelli 8, I-00168 Rome, Italy; 2Unicamillus, Saint Camillus International University of Medical Sciences, via di S. Alessandro 10, I-00131 Rome, Italy; 3Department of Systems Medicine, Medical Oncology Unit, University of Rome Tor Vergata, via Montpellier 1, I-00133 Rome, Italy; 4Endocrinology Unit, P.O.-S. Spirito in Sassia, ASL Roma 1, Lungotevere in Sassia 1, I-00193 Rome, Italy

**Keywords:** renal cell carcinoma, tyrosine kinase inhibitors, immune checkpoint inhibitors, bone metabolism, PTH, hypophosphatemia, hypocalcemia, osteonecrosis of the jaw

## Abstract

**Simple Summary:**

Tyrosine kinase inhibitors and immune checkpoint inhibitors have substantially prolonged survival in patients affected by advanced renal cell carcinoma. However, their impact on bone metabolism is less studied than in other malignancies. Nevertheless, the bone is the second site of metastatic spread in these patients, and chronic renal insufficiency, a condition associated with bone dysmetabolism, is frequently seen in patients who underwent nephrectomy due to renal cancer. Indeed, tyrosine kinase inhibitors may lead to hypophosphatemia and increased parathyroid hormone levels, with low–normal calcium levels, while immune checkpoint inhibitors may cause hypocalcemia and hypercalcemia. Remarkably, jaw osteonecrosis is a bone complication of tyrosine kinase inhibitors occurring more frequently in patients affected by renal cell carcinoma than other malignancies. The reasons for this prevalence are still unknown. In the literature, studies on epidemiology and pathogenetic mechanisms of bone dysmetabolism induced by those anticancer drugs are sparse. More robust studies are necessary to clarify these issues. In the meantime, Clinicians should consider the risk of bone dysmetabolism in patients affected by renal cell carcinoma on treatment with the agents mentioned above, particularly in those receiving antiresorptive medications (i.e., biphosphonates and denosumab), which can increase the risk of hypocalcemia and jaw osteonecrosis.

**Abstract:**

The medical therapy of advanced renal cell carcinoma (RCC) is based on the use of targeted therapies, such as tyrosine kinase inhibitors (TKI) and immune-checkpoint inhibitors (ICI). These therapies are characterized by multiple endocrine adverse events, but the effect on the bone is still less known. Relatively few case reports or small case series have been specifically focused on TKI and ICI effects on bone metabolism. However, the importance to consider these possible side effects is easily intuitable because the bone is one of the most frequent metastatic sites of RCC. Among TKI used in RCC, sunitinib and sorafenib can cause hypophosphatemia with increased PTH levels and low-normal serum calcium levels. Considering ICI, nivolumab and ipilimumab, which can be used in association in a combination strategy, are associated with an increased risk of hypocalcemia, mediated by an autoimmune mechanism targeted on the calcium-sensing receptor. A fearsome complication, reported for TKI and rarely for ICI, is osteonecrosis of the jaw. Awareness of these possible side effects makes a clinical evaluation of RCC patients on anticancer therapy mandatory, especially if associated with antiresorptive therapy such as bisphosphonates and denosumab, which can further increase the risk of these complications.

## 1. Introduction

Renal cell carcinoma (RCC) accounts for approximately 3% of all cancers, with the highest incidence occurring in Western countries and an annual increase in the last two decades of about 2% [1]. The RCC peak of incidence occurs between 60 and 70 years of age with a 1.5:1 predominance in men over women [2]. This year, in the United States, RCC is expected to be the sixth and ninth most frequently diagnosed malignancy in men and women, respectively [2].

RCC encompasses a heterogeneous group of malignancies deriving from the renal tubules’ epithelium. The most frequent histological subtypes include clear cell (ccRCC), accounting for approximately 75% of cases, and non-clear cell RCC (nccRCC), which includes papillary (types I and II; 10% of cases), chromophobe tumors (5%), and other rare forms (<5%) [3].

Several factors may contribute to the development of RCC, including diet, alcohol, smoking, obesity, poorly controlled hypertension, and environmental agents (i.e., cadmium, trichloroethylene, benzene, and asbestos) [4]. RCC may also be hereditary, e.g., Von Hippel-Lindau (VHL) disease, the most common condition associated with RCC [5,6].

RCC carcinogenesis is due to mutations in the genes of several molecular pathways, resulting in increased neo-angiogenesis, tumor heterogeneity, metabolic dysregulation, and detrimental tumor microenvironmental crosstalk, with hypoxia-induced factor (HIF) signaling pathway exerting a key role in these processes [7]. Indeed, in both sporadic (non-familial) and familial RCCs (e.g., VHL syndrome), the HIF transcription factors upregulate the expression of several growth factors (e.g., vascular endothelial growth factor [VEGF], platelet derived growth factor [PDGF], and transforming growth factor-alpha [TGF-α]), inducing angiogenesis, proliferation, and migration, as well as the expression of numerous genes regulating glucose metabolism and oxygen transport and metabolism [5,8,9,10,11,12].

Most RCC patients (approximately 75%) are diagnosed when the disease is localized and, thus, are candidates for radical nephrectomy with curative intent. However, advanced disease remains a major clinical issue, as approximately one-third of patients are diagnosed with locally advanced/inoperable or metastatic disease at their presentation, and in another 30% of patients, RCC recurs after radical nephrectomy [12]. The 5-year overall survival rate for all RCC subtypes is 49% [2], with recent significant survival improvement, mainly deriving from an increase in incidentally detected RCCs at earlier stages, better surgical techniques, and the availability of several new targeted and immunotherapy agents [4]. Notably, in 1990, the median survival of patients with advanced RCC was less than one year, with high-dose interleukin (IL)-2 and interferon-α the most effective agents (in a limited number of cases) [13], while it is currently over four years in patients receiving targeted therapies and immune checkpoint inhibitors (ICI) [14]. Moreover, some of these agents are recently approved as adjuvant treatment following radical nephrectomy [15].

Several anticancer agents are currently available for advanced RCC patients [16]. Tyrosine kinase inhibitors (TKI) targeting the vascular endothelial growth factor-receptor (VEGF-R) have represented the cornerstone of the management of advanced RCC for more than one decade. In contrast, although the inhibitors of the mammalian target of rapamycin (mTOR) have been considered an important therapeutic option in recent years as well, their use is now only restricted in the subsequent lines of therapy [17,18]. In this scenario, the advent of ICI has further revolutionized the treatment paradigm of RCC. Indeed, so far, several ICI (i.e., anti-cytotoxic T lymphocyte antigen-4 (CTLA-4), anti-programmed cell death protein-1 [PD-1], and anti-programmed death-ligand 1 (PD-L1)) have been approved for advanced RCC patients either as monotherapy or combined regimen [17,18].

Both targeted agents and ICI have substantially improved advanced RCC patients’ survival, but often at the cost of side effects that may worsen patients’ quality of life [19,20,21]. Although there are significant similarities and differences in terms of adverse events across the various classes of targeted agents, the toxicities in advanced RCC patients are generally manageable and commonly occur at the severity of grade 1 or 2 [22]. The most frequent reported adverse events are fatigue, cardiovascular (e.g., hypertension or bleeding/hemorrhage), gastrointestinal (e.g., diarrhea or mucositis), and endocrine (e.g., hypothyroidism) side effects [22,23,24]. In addition, targeted therapies are often responsible for dermatologic adverse events (e.g., hand-foot syndrome, skin rash, or desquamation) [25], which may require a dose reduction of the drug, prophylactic strategies, and supportive care [26]. In parallel, ICI are associated with immune-related adverse events (irAE), which are autoimmune conditions that can affect a broad range of organs [27]. Most frequently, irAE determine skin, hepatobiliary, gastrointestinal, endocrinological, and pulmonary toxicities in RCC patients [23,28,29]. Although irAE may vary in terms of onset and severity, they are often characterized by symptoms like those caused by targeted agents [30]. Therefore, early recognition of the specific drug responsible for the ongoing adverse event is essential to decide the best management for the patient, especially when they are treated with combined regimens [28].

In this scenario, it is undoubtedly that a deep knowledge of the safety profile of these anticancer agents is essential to decide the best therapeutic option for every patient affected by advanced RCC [28]. A large amount of literature is available on cardiovascular, endocrine, and gastrointestinal adverse events caused by the above-mentioned drugs in RCC patients. At the same time, data on their effect on the bone are sparse [24,31,32,33,34]. Herein, we reviewed the impact of targeted agents and ICI on bone and bone metabolism in patients affected by advanced RCC.

## 2. The Current Medical Treatment of Renal Cell Carcinoma: An Overview

Patients affected by advanced/inoperable or metastatic RCC may benefit from a therapeutic armamentarium that has continuously evolved in recent years, due to impressive improvements of overall survival demonstrated by targeted agents and ICI in large, randomized clinical trials (Table 1, Appendix A) [13,35,36,37,38,39,40,41,42,43,44,45,46,47,48].

Nowadays, the choice of first-line treatment in patients affected by advanced/inoperable or metastatic disease is based on the RCC subtypes (ccRCC versus nccRCC) and specific biochemical and clinical parameters included in either the “International Metastatic RCC Database Consortium” (IMDC) or “Memorial Sloan Kettering Cancer Center” risk groups [17]. Accordingly, the standard-of-care for the first-line treatment of ccRCC patients is represented by a combination regimen. Indeed, PD-1 inhibitor therapy with either VEGFR-targeted therapy or CTLA-4 inhibition has improved overall outcomes for patients with advanced ccRCC [49,50,51]. Nevertheless, while combinations of VEGFR-TKI with ICI can be administered irrespective of the IMDC risk group, the combination of ipilimumab (a CTLA-4 inhibitor) and nivolumab (PD-1 inhibitor) continues to be recommended only for patients with IMDC intermediate- and poor-risk disease [52]. In patients with contraindications to ICI, monotherapy with either sunitinib, pazopanib, tivozanib, or cabozantinib as the alternative to PD-1 inhibitor-based first-line combinations [17]. After progression on PD-1 inhibitor-based combination, it is recommended to sequence VEGFR-TKI therapy using other agents effective as first-line treatment but not administered previously. Conversely, the possibility to continue ICI after progressing on ICI-based combination is not recommended due to lack of data [17]. In patients affected by metastatic papillary RCC, monotherapy with cabozantinib is considered the preferred therapeutic option in the first-line setting [28]. In the case of progressive disease, second-line therapy should be based on other first-line agents not administered previously, such as sunitinib, pembrolizumab, and savolitinib (in MET driven RCC) [17].

## 3. Effects of TKI in Bone Metabolism

### 3.1. Molecular Target in Bone Metabolism

TKI actions on bone metabolisms are probably mediated by the inhibition of TKs expressed by the bone cells, such as PDGFRα, PDGFRβ, c-KIT and C-FMS (Figure 1). PDGF is involved in bone metabolism because it stimulates osteoclastogenesis [53] and subsequent bone resorption, through the production of several cytokines, including receptor activator of nuclear factor kB ligand (RANKL) and macrophage colony stimulating factor (M-CSF) and IL-6. C-kit binds to ligand stem cell factor (SCF), which stimulates osteoclast precursor proliferation [54], and transduces along the mitogenic pathways of microphthalmia transcription factor (MitF). MitF, in turn, encodes for a transcription factor acting downstream of the RANK/RANKL pathway [55]. *MITF* deficiency is associated with a form of autosomal recessive osteopetrosis in the clinical features of the COMMAD (coloboma, osteopetrosis, microphthalmia, macrocephaly, albinism, and deafness) syndrome [56]. SRC kinase stimulates osteoclastogenesis, osteoclastic activity, and survival, leading to increased bone resorption, as demonstrated by mouse models carrying a targeted disruption of the gene *Src*, that develop osteopetrosis [57]. Macrophage colony-stimulating factor (M-CSF) is also secreted by osteoblasts upon PTH stimulation, and has a key role in osteoclast proliferation and differentiation [58]. VEGF is one of the most important growth factors for regulation of vascular development and angiogenesis and it is often targeted in several subtypes of cancer in order to reduce tumor vascularization. Bone is a highly vascularized tissue and angiogenesis plays an important role in osteogenesis [59]. Therefore, VEGF is crucial in skeletal development, mediating blood vessel formation and the vascularization of cartilage into bone. VEFG receptors are present in both osteoblasts and osteoclasts [60].

### 3.2. Effect of TKI on Bone Metabolism: A Lesson from Imatinib-Treated Patients

Alterations in bone metabolism during TKI therapy were first reported for imatinib. Imatinib mesilate is an orally active TKI with activity against several tyrosine kinases, such as Abl, Arg (Abl-related gene), c-KIT, PDGF-R, and BCR-ABL. Imatinib is used in patients with chronic myeloid leukemia (CML) and malignant gastrointestinal stromal tumors (GIST) [61].

In some patients enrolled in clinical trials and treated with imatinib for newly diagnosed CML, hypophosphatemia (serum phosphate levels <2.5 mg/dL) has been observed [62]. Berman et al. reported hypophosphatemia in 16 out of 24 patients who were receiving imatinib. In this group, hypophosphatemia was associated with elevated PTH levels, low-to-normal serum calcium levels, and decreased biochemical markers of both bone formation and resorption. Interestingly, they received a higher dose of imatinib than patients in the normal-phosphate group [63]. Subsequent studies confirmed that the use of imatinib is associated with a decrease in calcium and phosphate, secondary hyperparathyroidism, a transient increase in bone formation markers, and a decrease in bone resorption markers [64]. 1,25-hydroxyvitamin D3 resulted in an increase, probably as a consequence of PTH increases [63].

Considering that the homeostatic control of serum phosphate and calcium is obtained through the resorption of phosphate and calcium by the kidneys, the absorption of dietary phosphate and calcium by the gut, and through the dissolution of phosphate and calcium from bone, probably the genesis of hypophosphatemia observed in imatinib-treated patients is related to one or more of these mechanisms [65]. It is reasonable to consider that the decreased intestinal absorption of phosphate and calcium can be secondary to gastrointestinal side effects such as diarrhea, affecting about 24% of imatinib-treated patients [66]. The reduction in calcium levels could represent a stimulus for the increased PTH secretion by the parathyroid glands and, in turn, the secondary increase of PTH can mediate the increased phosphaturia and the alteration in phosphate reabsorption, which causes hypophosphatemia [65]. PTH also stimulates the production of 1,25-dihydroxyvitamin D3 through the activation of the 1-alphahydroxylase enzyme. This active form of vitamin D should stimulate phosphate and calcium reabsorption by the gut and reduce the phosphate excretion from the kidney. However, in imatinib-treated patients, hyperphosphaturia is observed, together with increased active vitamin D levels, probably because the hyperphosphaturic action of the PTH overrides the inhibition of phosphate excretion mediated by vitamin D.

Another possible mechanism explaining the decreased serum calcium and phosphate levels could be related to their sequestration in the bone, caused by a decrease in bone resorption and/or an increase in bone deposition. In fact, patients undergoing imatinib therapy experience dysregulated bone remodeling. Imatinib treatment results in skeletal side effects, leading to increased bone volume and density. Most of the clinical studies have been performed on patients with CML. The trabecular bone volume (TBV) in iliac crest bone biopsies obtained from CML patients substantially increased (>2-fold) after 2 to 4 years of imatinib therapy in about 50% of patients [67]. Another study showed a small increase in the lumbar spine and total hip bone mineral density (BMD) in a group of patients treated with long-term imatinib therapy [68]. The most accepted theory is that the drug decreases osteoclastic bone resorption. The inhibition of osteoclast activity, if not associated with a decrease in osteoblast activity, could explain the role of imatinib in the increase in bone volume. In vitro studies demonstrated that imatinib inhibits the differentiation and the activity of osteoclasts and inhibits the survival of osteoclast precursors. On the other hand, osteoblast activity may be transiently increased by imatinib treatment, but subsequently, it decreases to baseline levels or lower [65].

In a group of 11 patients treated with imatinib at a high dose (600 mg/day), an increase in trabecular bone volume and trabecular thickness has been reported, associated with a significant decrease in osteoclast numbers at iliac crest biopsy and with a significant decrease in serum levels of a marker of osteoclast activity, while the osteoblast number did not change [76]. The inhibition in osteoclastogenesis opens the potential therapeutic role of imatinib as an antiosteolytic agent in diseases such as osteoporosis, multiple myeloma, and metastatic bone disease [77]. In the latter groups of patients, the possibility of combining both the antineoplastic and antiosteolytic activity in a single agent is particularly interesting. However, it is important to note that imatinib caused a site-specific decrease in BMD at the femoral neck [76], even if in a small group of patients. Hip fractures, as well known, are an important cause of disability and mortality, and they are strongly associated with decreased BMD. Therefore, this specific side effect of imatinib should be considered and further carefully evaluated by more robust studies.

### 3.3. Effects on Bone of TKI Used for Renal Cell Carcinoma

Sunitinib is a TKI targeting different kinases (VEGFR1–3, PDGFRα and β, FLT3, KIT, CSR, and RET). The mechanism of action of sunitinib is mainly based on the blocking of angiogenesis through VEGF-signaling disruption. Few effects of sunitinib on bone have been reported. As for imatinib, a reduction in serum phosphate levels has been reported in 13% of patients receiving sunitinib [78]. In 2011, Mazzanti reported the onset of primary hyperparathyroidism in a cohort of patients affected by mRCC and treated with sunitinib [79]. In particular, PTH and calcium metabolism evaluation have been prospectively performed in 25 patients receiving sunitinib 50 mg/day. During the treatment period, 68% of patients developed elevated serum PTH with low-to-normal serum calcium and phosphate. In the group with an elevation of PTH, low or undetectable urinary calcium levels have been reported [79]. A few months later, Baldazzi and colleagues reported an increase in PTH levels compared with baseline in 18 out of 26 patients in therapy with sunitinib for mRCC, with stable serum calcium levels [80]. Once the PTH level increased, it resulted stable during sunitinib treatment. In patients who developed hyperparathormonemia and sunitinib was discontinued because of disease progression or toxicity, PTH levels progressively returned to the normal range within 2 to 4 months after sunitinib withdrawal, suggesting a causal relationship between the drug and the alterations of PTH levels [80]. In this group of patients, increased 1,25-dihydroxyvitamin D3 levels were observed, according to the increase in PTH levels, while mean urinary calcium was reduced. The mechanisms at the basis of these findings are still unknown, but different hypotheses can be supposed. A possible explanation for the PTH level increase could be related to the decrease of serum ionized calcium during sunitinib treatment caused by intestinal malabsorption. Another possible explanation is that sunitinib may inhibit the action of PDGFR on both osteoclasts and osteoblasts [60]. While data about the effects on osteoblasts are scarce, the inhibitory action on osteoclasts has been documented for sunitinib. In bone-metastatic mouse models obtained by the transplantation of luciferase-labeled ACHN (Luc) RCC cells, sunitinib prevented the growth of metastasis, and the number of osteoclasts was significantly lower in sunitinib-treated mice compared with controls [81]. In the same paper, the authors evaluated the levels of serum and urinary amino-terminal telopeptide (NTx), a marker of bone resorption, in 16 patients with mRCC treated with sunitinib. During the first 4 weeks of treatment, serum and urinary NTx significantly decreased [81].

Sorafenib inhibits RAF/MAPK pathway, VEGFR 2 and 3, PDGFR, and c-Kit [82]. Hypophosphatemia has been reported as a common side-effect, occurring in about 45% of patients [83], but the involved mechanism is still unclear. Bellini and colleagues reported the case of a 64 years-old patient with mRCC treated with sorafenib (400 mg b.i.d. orally), who developed hypophosphatemia after 1 month of treatment associated with a decrease in serum calcium and elevated PTH levels [84]. In this case, FGF23 and vitamin D levels were both decreased, suggesting that the pathogenesis of hypophosphatemia was related to reduced intestinal phosphate, calcium, and vitamin D absorption, while the role of FGF23 was not contributory. In this case, concomitant systemic malabsorption has been ruled out since no diarrhea was documented [84]. On the other hand, the normalization of phosphate levels after cholecalciferol supplementation suggests the utility of treating these patients with vitamin D to avoid this common metabolic disorder. Sorafenib inhibited osteoclast activity, as documented by the decrease of serum CTX [84]. This effect has also been noted in a group of patients after 2 weeks of sunitinib treatment [85]. Furthermore, both sunitinib and sorafenib have been shown to reduce the bone resorption marker (urinary N-telopeptide) in mRCC patients with bone metastasis [86] but, even if studies of comparison are lacking [60], their effect seems to be lower than that of other anti-resorptive agents such as bisphoshponates (BP) or denosumab. Unfortunately, despite this “protective” effect on the bone, sorafenib use is associated with sarcopenia in patients with mRCC [87], which represents a risk factor for falls and, consequently, for fragility fractures.

Pazopanib and axitinib are second-generation TKIs targeting multiple proteins involved in tumor cell proliferation and angiogenesis. Axitinib is a more selective TKI, with in vitro potency higher than that of pazopanib and of the first-generation VEGFR inhibitors such as sunitinib and sorafenib. Furthermore, its more specific therapeutic window could cause fewer off-target adverse effects [88]. Few data are available concerning the bone effects of these two compounds. In a group of 24 patients treated with neoadjuvant axitinib for locally advanced RCC, the study of body composition showed a decrease in skeletal muscle volume as well as weight loss and sarcopenia [89].

Cabozantinib, a nonselective RET inhibitor additionally targeting VEGFR2 and MET, is currently approved for mRCC treatment in both the first and second lines. The theoretical effects of cabozantinib on the bone microenvironment can be supposed by the action of cabozantinib on the hepatocyte growth factor (HGF)/MET pathway, which has been shown to affect osteopontin expression in osteoblasts [90]. In a preclinical study, doses of cabozantinib below the threshold of cytotoxicity inhibited osteoclast differentiation and bone resorption activity [91]. Additionally, cabozantinib downregulated the expression of osteoclast marker genes (TRAP, cathepsin K, and RANK). The treatment did not have direct effects on osteoblast viability or differentiation, but it increased osteoprotegerin mRNA and protein levels. Furthermore, the cell-to-cell contact between cabozantinib pre-treated osteoblasts and untreated osteoclasts showed an indirect anti-resorptive effect of cabozantinib [91]. An Italian prospective study involving 39 patients with mRCC treated with cabozantinib, showed a reduction of CTx and N-terminal propeptide of type 1 collagen (PINP) levels, with a transitory increase of PTH after 3 months of treatment. However, these changes in bone metabolism markers are not associated with the clinical outcome [92].

### 3.4. Jaw Osteonecrosis Associated with TKI Use

Jaw osteonecrosis (ONJ) is a rare but potentially severe condition that can affect both jaws, although it is more common in the mandible. It manifests as one or more necrotic bone lesions in the oral cavity, which persist for at least 8 weeks in patients receiving an antiresorptive medication for primary or metastatic bone cancer, osteoporosis, or Paget’s disease, without a history of radiation therapy to the jaws [93].

The first case of ONJ related to the use of BP was published in 2003 [94], and since then, the reports concerning this condition have increased. The term “bisphosphonate-related osteonecrosis of the jaw” (BRONJ) was updated by the AAOMS in 2014 and 2015 [95], with the recommendation to change the nomenclature in “medication-related osteonecrosis of the jaw” (MRONJ). This change has been considered mandatory for the growing number of osteonecrosis cases involving the maxilla and mandible associated with other antiresorptive (denosumab) or anti-angiogenic therapies. In fact, TKIs have been described to be responsible for ONJ, both alone and in combination with antiresorptive medication. The possible etiology has been referred to as the intrinsic risk of ONJ related to the use of TKIs, as confirmed by several papers published in the last few years [96,97,98]. However, in many cases, it is not possible to quantify the risk or to fully define the etiopathogenic relationship. In general, the ONJ associated with TKI use may appear spontaneously or after surgery and is more common in the mandible. The time of onset of ONJ after starting the TKI use may be variable (from a few weeks to 15 months) [99].

The occurrence of this rare but fearsome complication should be carefully considered and evaluated in RCC patients. In fact, after the lung, the bone represents the second most common metastatic site in advanced RCC [100]. Bone metastases are associated with shorter survival because the lesions are often osteolytic, and they can lead to skeletal-related events, such as pathologic fractures, spinal cord compression, and hypercalcemia [101]. The occurrence of skeletal complications can be prevented by the use of BP and denosumab, which are well-known to be associated with hypocalcemia and ONJ. Recent evidence has confirmed a relatively high ONJ risk in patients with a combined administration of BP and targeted drugs. Within the RCC population, a higher incidence of ONJ has been noted in patients treated with TKI (sorafenib, sunitinib, and axitinib in one case), plus intravenous BP compared with the use of BP alone [60]. Renal cell cancer patients are a population “at risk” probably because of the enlarged administration of targeted therapies [102]. Based on data from the analysis of 3 randomized trials comparing zoledronic acid and denosumab, the estimated frequency by “cancer type” of ONJ is 3.9% for RCC, more than twice the ONJ rate of the whole patient population (1.6%) [103].

In a retrospective study on RCC patients affected by bone metastases, including 76 patients, the concomitant use of BP and TKI, which has been proven to improve the median progression-free survival and the median overall survival, was associated with a 10% incidence of ONJ [104].

A multicenter retrospective study involving nine Italian centers and 44 patients evaluated the clinical features of RCC patients treated with BP and anti-angiogenic agents who developed ONJ. Patients were mainly male (82%) and the median age was 63 years. The most used drugs were zoledronic acid (93%) for an antiresorptive agent and sunitinib (80%) as anti-angiogenic therapy, but also patients on pamidronate, ibandronate, sorafenib, bevacizumab, mammalian target of rapamycin inhibitors have been included. The mandible was the most involved site (52%); both jaws were affected in 12% of cases, while the most common precipitating events were dental/periodontal infection (34%) and tooth extraction (30%) [105]. Another multicenter retrospective study involving 41 patients evaluated the toxicity profile in terms of the occurrence of ONJ and hypocalcemia in patients with mRCC treated with denosumab and anti-angiogenic therapy. In this group, 7 patients (17%) developed ONJ (6 of them received sunitinib as anti-angiogenetic therapy), with a median time exposure before ONJ of 19.9 months [106].

A pharmacovigilance study published in 2021 evaluated the possible association between lenvatinib and ONJ through the analysis of the “FDA adverse events reporting system”, an efficient tool designed by the FDA to support post-marketing safety surveillance. The results showed a low rate (0.5%) of ONJ in patients treated with lenvatinib alone. In contrast, a higher incidence of ONJ was documented in patients with other concomitant drugs (glucocorticoids, anti-resorptives, and anti-angiogenics) or risk factors (dental procedures or diabetes mellitus) [107].

Single case reports of RCC patients treated with TKI who developed ONJ have been reported in Table 2.

Despite the limitations deriving from the retrospective studies and from the single case reports, the high rate of ONJ in RCC patients treated with TKI alone or in combination with antiresorptive agents makes it mandatory to consider this potential adverse effect. Even if the full etiology of ONJ is unknown, a periodical dental examination and patient education in oral health is crucial to reduce ONJ occurrence. 

## 4. Effects on Bone of ICI Used for Renal Cell Carcinoma

The clinical use of the ICI is associated with multiple endocrine adverse events, which have been widely described [31], but the effects on skeleton and bone metabolism are less studied. However, studies concerning other diseases (rheumatoid arthritis, postmenopausal osteoporosis, periodontal disease, and HIV) [117,118] confirm a direct relationship between the immune system and bone metabolism. Pro-inflammatory cytokines, particularly tumor necrosis factor (TNF) and interleukins IL-1, IL-6, and IL-17, play a crucial role in the pathogenesis of inflammation-induced bone loss. The TNF-alpha stimulates the receptor activator of nuclear factor kappa B ligand (RANKL), inducing osteoclast maturation and activation and inhibiting osteoblast function [119]. In mice, the absence of IL-1 protects from bone loss, suggesting that IL-1 is an essential mediator of inflammatory osteopenia [120]. IL-6 binds receptors on pre-osteoclasts, promoting osteoclastogenesis and resulting in increased levels of bone resorption [121]. Furthermore, IL-6 can promote PTHrP secretion via TNF-alpha, with enhanced osteoclastogenesis, bone loss, and hypercalcaemia [122]. ICI therapy promotes a pro-inflammatory state with similar mechanisms, activating T-cells, which in turn secrete cytokines with antineoplastic effects, such as TNF-alpha, IL-1, IL-4, IL-6, IL-17, and INF-gamma [69] (Figure 1).

Despite this pathogenetic background, few data are available concerning the effects of ICI on bone metabolism, especially in patients treated for mRCC. In the majority of cases, data derives from studies on other malignant diseases that were treated with the same therapeutic schemes used for mRCC.

From a clinical point of view, the risk of ICI-associated osteoporosis represents the most relevant problem, especially in mRCC patients. In fact, as above-mentioned, bone metastases are often present in advanced disease, and osteolytic lesions represent per se an independent risk factor of fractures. A small case series reported in 6 patients treated with anti-PD-1 or anti-PD-1/CTLA-4 therapy, the onset of fractures (n. 3) and resorptive bone lesions (n. 3). In this group of subjects, only one patient, treated with nivolumab, was affected by RCC. However, the other patients, affected by different malignancies, had been treated with therapeutic schemes usually used for the treatment of mRCC (pembrolizumab, nivolumab, or ipilimumab plus nivolumab) [70].

ONJ, which represents a fearsome complication related to the use of TKI, has been reported in two patients treated with ipilimumab [71] and nivolumab [72], in both cases for advanced melanoma. Recently a case of peri-implantitis-induced MRONJ involving the right upper jaw has been reported in an mccRCC patient treated with multiple therapies (interleukin-2, bevacizumab, zoledronic acid, denosumab, nivolumab, and cabozantinib) [73].

Clinical trials, such as KEYNOTE-189 [74] and CHECKMATE-067 [75], have reported a 21–29% hypocalcemia event rate as an ICI side effect, and several case reports have been published. In their retrospective study, Nalluru and colleauges evaluated the incidence of hypocalcemia in 178 patients on ICI treatment (pembrolizumab, nivolumab, ipilimumab, and nivolumab/ipilimumab). Eighteen patients were affected by RCC. Hypocalcemia events were 8.4% in patients with no calcium-altering conditions or medications and 19.3% in the presence of calcium-altering factors (calcium supplements, vitamin D, BP, chronic kidney disease, and bone metastasis). The reduced incidence in this study was probably due to the definition of “true hypocalcemia”, which was calculated after correcting calcium for albumin from the initiation of ICI to their last follow-up. The percentage of patients with hypocalcemia was much higher and similar to the KEYNOTE-189 and CHECKMATE-067 trials when serum calcium values without albumin correction were calculated [123].

The occurrence of hypocalcemia during treatment with ipilimumab/nivolumab for metastatic melanoma has been reported by other authors. In one case, hypocalcemia has been associated with undetectable PTH levels, suggesting direct damage to the parathyroid glands that required long-term treatment with calcium plus vitamin D [124]. After this report, a few years later, Piranavan and colleauges described another case of severe hypocalcemia (5.8 mg/dL) associated with low PTH (7.77 pg/mL) in a 61-year-old female patient treated with nivolumab for metastatic small cell lung cancer [125]. Interestingly, considering autoimmunity as an important cause of hypoparathyroidism, the evaluation of calcium-sensing receptor (CaSR)-activating autoantibodies has been performed, resulting in elevated autoantibodies levels. The activation of CaSR by autoantibodies cause reduced PTH secretion and hypocalcemia, with similar mechanisms reported for the calcimimetic drug cinacalcet [126]. The temporal relationship between nivolumab administration and hypocalcemia suggests a strong causal relationship. Several other authors reported hypocalcemia secondary to ICI treatment related to a possible autoimmune mechanism that can be caused by either immune-mediated destruction or by the hyperactivation of the CaSR by activating autoantibodies, as previously reported. Dadu and colleagues reported the case of a 73-year-old man who developed severe symptomatic hypocalcemia after the initiation of ipilimumab plus nivolumab for the treatment of metastatic melanoma. Pre-existing autoimmune hypoparathyroidism was ruled out, while the detection of CaSR-activating antibodies after ICI treatment supports the development of autoimmune mechanisms induced by the drugs [127].

In the patient described by Trinh [128], affected by advanced melanoma treated with ipilimumab + nivolumab, the hypoparathyroidism was responsive to immune-suppressive medication, confirming the autoimmune-mediated mechanism. The patient reported by El Kawkgi developed hypocalcemia with low PTH levels during ipilimumab plus nivolumab treatment, but CaSR autoantibodies have not been measured [123].

On the other hand, cases of ICI-related hypercalcemia have also been reported. Recently, Johnson and colleagues described a 68 years-old mRCC patient who developed hypercalcemia (12.8 mg/dL) immediately after two cycles of nivolumab and ipilimumab [129]. Elevated calcitriol level (142 pg/mL) and low parathyroid hormone (PTH) and parathyroid hormone-related protein (PTHrP) have been detected, suggesting that the hypercalcemia has been mediated by the elevation of calcitriol levels. Interestingly, the elevated calcium levels were resistant to treatment with calcitonin, hydration, and zoledronic acid but responded to high-dose prednisone (1 mg/kg), suggesting an autoimmune genesis. In particular, the authors suggest a calcitriol-induced hypercalcemia related to the inflammatory condition mediated by macrophage activation. They speculate that the PD-1/PD-L1 blockade can activate macrophages and T cells, which in turn express increased levels of 25(OH) D-hydroxylase. Both macrophages and T cell activation lead to increased conversion of 25(OH) vitamin D to calcitriol [129]. However, considering the rareness of the reported cases, a more robust clinical experience is mandatory to establish the pathogenetic mechanisms.

## 5. Discussion

The recent advances in genomics and molecular biology, along with the clarification of the cancer-immunity cycle, have profoundly changed the management of advanced RCC patients [130]. In the last two decades, the treatment paradigm of RCC has evolved from immune cytokine-based approaches to targeted therapies consisting of TKI against VEGF-R (and other key molecules in angiogenic pathways) and inhibitors of mTOR [13]. Moreover, the recent advent of ICI has represented a novel breakthrough in the immunotherapy of advanced RCC [130].

In this scenario, several studies aimed to evaluate the safety profile of these anticancer agents, investigating the incidence of the different toxicities and their best management [22]. Despite significant similarities and differences across the various classes of targeted agents, the related adverse events in advanced RCC patients are frequent but generally manageable, commonly occurring at the severity of grade 1 or 2 [22,131]. The most commonly reported adverse events are fatigue, hypertension, bleeding/hemorrhage, hypothyroidism, diarrhea, elevated liver enzymes, skin rash, and hand-foot syndrome [131]. Although close vigilance and prompt detection are mandatory for an efficient resolution, an intervention with dose reduction and/or supportive care is essential to manage adverse events. In parallel, ICI has also shown manageable safety profiles in advanced RCC patients, although they are associated with a spectrum of irAE, sometimes severe but rarely fatal [26]. The management of irAE can be complex, but corticosteroids generally are the treatment’s mainstay. However, some irAE may also require immunosuppressive therapy with mycophenolate mofetil or other agents [18].

A large amount of literature is currently available on the skin, cardiovascular, gastrointestinal, and endocrine adverse events triggered by the above-mentioned drugs in RCC patients [18,22,30,131]. However, data on their effect on the bone are sparse, and thus, we reviewed the available literature in this direction.

In most cases, studies concerning this topic were case reports or retrospective evaluations. Moreover, the patients evaluated were treated with TKI or ICI, due to different types of cancer, including RCC. Therefore, specific guidelines or recommendations about clinical management in this setting are lacking.

A reasonable approach is to consider the possibility of altered bone metabolism on TKI and ICI treatment and to perform specific clinical and laboratory evaluations, possibly in the context of a multidisciplinary approach with endocrinologists. Hypocalcemia and hypercalcemia, as well as hypophosphatemia, can represent life-threatening conditions; it is, therefore, helpful to check serum calcium and phosphorus before and during anticancer treatment. In the presence of hypocalcemia, treatment with intravenous calcium gluconate and oral calcitriol is suggested [123]. Calcitriol, the hormonally active synthetic vitamin D, represents the therapeutic gold standard of hypocalcemia in patients with chronic kidney disease. Some authors suggest that patients receiving imatinib ingest an amount of calcium and vitamin D (as cholecalciferol) from diet and/or supplements because this may help reduce secondary hyperparathyroidism and prevent hypocalcemia [60]. Hypercalcemia must be treated with BP, as zoledronic acid or pamidronic acid are associated with hydration. In cases of non-responsive to bisphosphonate treatment, prednisone should be used, considering that anticancer-related hypercalcemia could be mediated by an autoimmune mechanism [129]. A periodical evaluation of bone mineral density by dual-energy X-ray absorptiometry must be provided, and the fracture risk should be evaluated for each patient in a multidisciplinary approach. Based on these evaluations, treatment with oral calcium/cholecalciferol supplementation and antiresorptive agents must be considered. However, it is mandatory to consider the possible occurrence of osteolytic bone metastases in RCC, since hematogenous spread to the bone is frequent in these patients [132], leading to an increased risk of skeletal complications. The anti-osteoporotic BP (especially zoledronic acid) and denosumab are often used to reduce the adverse skeletal reaction and worsening pain associated with bone metastases, regardless of BMD levels. Careful dental care is mandatory to prevent or mitigate the occurrence of jaw osteonecrosis, especially in patients treated with concomitant anti-resorptive therapy or in the presence of other risk factors [133]. The current evidence-based guidelines recommend regular dental examinations before and during treatment, stabilization of oral disease before starting therapy, and maintenance of good oral hygiene [134].

In our opinion, since specific studies are unavailable, patients affected by RCC would deserve a better evaluation of the “bone effects” of ICI and TKI due to various peculiar aspects. First, nephrectomy-associated chronic kidney disease represents per se a risk factor for bone dysmetabolism. Furthermore, around one-third of patients with metastatic RCC present with bone metastases at the diagnosis [135]. After the introduction of targeted therapy and the extended overall survival, the rate of skeletal-related events has become even more prevalent [136]. Considering the paucity of available data, it is unknown if the incidence of bone effects in RCC patients treated with TKI and/or ICI is higher than in patients treated for other cancers. For the considerations mentioned above, we think this topic should be thorough by focused studies.

Finally, recent evidence suggests that some agents, i.e., everolimus, an mTOR inhibitor active in the treatment of RCC, might not damage bone metabolism. In fact, data deriving from its use in breast carcinoma confirmed a “bone protective” effect on bone. The mTOR pathway interacts with multiple endocrine signaling pathways in breast cancer cells, such as the estrogen receptor pathway and other growth factors involved in tumor progression and resistance to endocrine therapy [137]. The inhibition of the mTOR pathway with everolimus may enhance sensitivity to endocrine therapy and exert a direct protective effect on bone [137]. In vitro studies performed in a human co-culture model used to mimic the crosstalk between Caki-2 cells (ccRCC) and osteoclasts demonstrated that the sequential combined treatment of everolimus and zoledronic acid is the most effective in the inhibition of both Caki-2 cells survival and osteoclastogenic potential, making it an effective strategy to inhibit bone metastasis [138]. However, clinical evidence concerning the effect of everolimus on bone metabolism in RCC patients is lacking. This is probably due to the difficulty of obtaining long-term data: everolimus is currently used as second or third-line therapy in patients with advanced RCC, and the limited PFS of these patients makes it very difficult to obtain an adequate evaluation of the drug’s impact on bone metabolism. However, despite specific limits, these studies highlight the possibility that experimental models might help clarifying the mechanisms underlying the effects on the bone of more largely used drugs (i.e., TKI and ICI, as single agents and in combination) for the treatment of advanced RCC, along with future agents.

## 6. Conclusions

RCC is characterized by a poor prognosis in advanced stages. TKIs and ICI, which represent the mainstay of malignancy treatment at this stage, are associated with different endocrine side effects. Despite the growing interest in the endocrine adverse events of these anticancer drugs, the effect on the bone is less studied. However, the prevalence of metastases to the bone, representing the second metastatic site of advanced RCC, suggests that particular attention should be paid to the effects of TKI and ICI on bone metabolism in these patients. The most important effects on bone metabolisms related to the use of TKIs in RCC are hypophosphatemia and increased PTH levels, with low–normal calcium levels, but a fearsome bone complication is ONJ, which is directly associated with the use of TKIs: this adverse event showed higher prevalence in patients affected by RCC than other malignancies. On the other hand, most of the ICI-related side effects are mediated by autoimmune mechanisms: both hypocalcemia and hypercalcemia have been reported, but the clear pathogenetic mechanism is still far to be fully elucidated. More robust studies are necessary to identify the prevalence of bone metabolism-related side effects in RCC patients receiving the current systemic anticancer treatments and possible strategies to prevent them. Since bone metabolism is regulated by multifactorial agents, a new frontier is the influence of gut microbiota on bone health [139,140]. TKI and ICI are known to alter the microbiota [141]. Therefore, a combined assessment of these factors is a promising topic. In the meantime, clinicians should consider the risk of the above-mentioned side effects, particularly in RCC patients on treatment with TKIs and ICI. This is even more important for patients receiving these anticancer therapies in association with antiresorptive medications (such as BP and denosumab), which can increase the risk of potential complications such as acutely hypocalcemia and, over long term, ONJ.

## Figures and Tables

**Figure 1 cancers-15-00529-f001:**
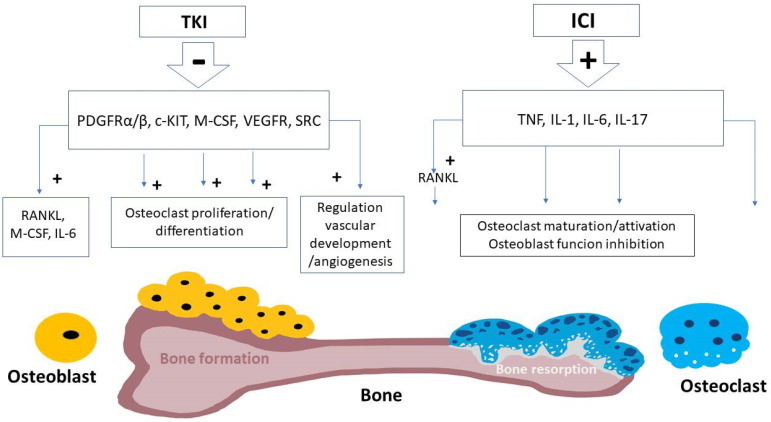
Schematic mechanisms of bone remodeling induced by tyrosine kinase inhibitors and immune checkpoint inhibitors. Tyrosine kinase inhibitors (TKI) cause the inhibition of various receptor kinases of bone cells, negatively affecting osteoclast proliferation/differentiation and regulation of vascular development and angiogenesis (see text) [61,62,63,64,65,66,67,68]. Immune checkpoint inhibitors (ICI) are suggested to mediate their bone effects, in terms of bone resorption, through the stimulation of pro-inflammatory cytokines, including tumor necrosis factor (TNF) and various interleukins (i.e., IL-1, IL-6, and IL-17) [69,70,71,72,73,74,75].

**Table 1 cancers-15-00529-t001:** List of the main anticancer agents available for RCC treatment as monotherapy and/or combined regimens.

Vascular Endothelial Growth Factor Receptors (VEGFR) TKI		
*Name*	*Targets*	*Inhibitor Class*
Axitinib	VEGFR 1-3, PDGFRβ	II
Cabozantinib	RET, MET, VEGFR1-3, KIT, TrkB, FLT-3, AXL, TIE-2, ROS1	I
Lenvatinib	VEGFRs, FGFRs, PDGFR, KIT, RET	II
Pazopanib	VEGFR1/2/3, PDGFRα/β, FGFR1/3, KIT, LCK, FMS, ITK	I
Sunitinib	PDGFRα/β, VEGFR1-3, KIT, FLT-3, CSF-1R, RET	II
Tivozanib	VEGFR 1-3, KIT	-
	**Mammalian Target of Rapamycin (mTOR) Inhibitors**	
*Name*	*Target*	
Everolimus	mTOR	
**Immune Checkpoint Inhibitors**		
*Name*	*Target*	*IgG class*
Avelumab	PD-L1	IgG1
Ipilimumab	CTLA-4	IgG1
Nivolumab	PD-1	IgG4
Pembrolizumab	PD-1	IgG4
**Combined regimens**		
*Drugs*	*Clinical Trial*	*FDA Approval*
Lenvatinib + Everolimus	NCT01136733	2016
Nivolumab + Ipilimumab	CheckMate 214	2018
Pembrolizumab + Axitinib	Keynote 426	2019
Avelumab + Axitinib	JAVELIN Renal 101	2019
Nivolumab + Cabozantinib	CheckMate 9ER	2021
Pembrolizumab + Lenvatinib	CLEAR	2021

Abbreviations: cytotoxic T-lymphocyte associated protein 4 (CTLA-4); colony-stimulating factor 1 receptor (CSF-1R); fibroblast growth factor receptors (FGFR); Fms-like tyrosine kinase 3 (FLT3); IL2 inducible T cell kinase (ITK); lymphocyte-specific protein tyrosine kinase (LCK); programmed Cell death protein 1 (PD-1); platelet-derived growth factor receptor (PDGFR); programmed death-ligand 1 (PD-L1); endothelial-enriched tunica interna endothelial cell kinase 2 (TIE2); tropomyosin receptor kinase B (TrkB).

**Table 2 cancers-15-00529-t002:** Case reports of ONJ in RCC patients treated with TKI or TKI + BP.

Drug	TKI Alone	TKI + BP	Number of Cases	Year	Reference
Sunitinib		x	3 patients (2 had mucositis, which has been considered as a risk factor of ONJ)	2010	Hoefert [108]
Sunitinib		x	1 patient	2010	Bozas [109]
Sunitinib	x		1 patient	2011	Koch [110]
Sunitinib	x (1 case associated with cisplatinum)		2 patients	2012	Nicolatou-Galitis [111]
Sunitinib	x		1 patient	2012	Fleissig [112]
Sunitinib		x	2 patients	2012	Agrillo [113]
Axitinib	x		1 patient	2017	Patel [98]
Sunitinib	x		1 patient	2017	Ashrafi [114]
Lenvatinib	x		1 patient	2019	Mauceri [115]
Lenvatinib	x		1 patient	2021	Monteiro [116]

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
