# Peer review of "Bone Metabolism Effects of Medical Therapy in Advanced Renal Cell Carcinoma"

_cancers, 2023, doi:10.3390/cancers15020529_

Round 1
Reviewer 1 Report
The review article by Paragliola R.M. et al. have systematically studied the effect of tyrosine kinase inhibitors (TKI) and immune-checkpoint inhibitors (ICI) on bone metabolism in renal cell carcinoma (RCC) patients. The manuscript is well written with clear description as well as proper discussion. However, some concerns should be addressed.
1. The interval time of references in the introduction is too long and contains a maximum of older references, so it is suggested to quote the literatures in the last three to five years.
2. Authors have described more about TKI, so it will be great to expand the effects of ICI in bone metabolism used for renal cell carcinoma.
3. In the discussion part, it is suggested to add and explain, what could be an alternate to overcome hypophosphatemia, hypocalcemia, and hypercalcemia in RCC patients after treating with TKI and ICI.
4. The language expression of the manuscript needs to be further simplified and polished.
Author Response
|
Reviewer #1
The review article by Paragliola R.M. et al. have systematically studied the effect of tyrosine kinase inhibitors (TKI) and immune-checkpoint inhibitors (ICI) on bone metabolism in renal cell carcinoma (RCC) patients. The manuscript is well written with clear description as well as proper discussion. However, some concerns should be addressed.
|
|
|
Observations/suggestions |
Authors’ answers |
|
The interval time of references in the introduction is too long and contains a maximum of older references, so it is suggested to quote the literatures in the last three to five years. |
We thank the Reviewer for his/her general comments concerning our manuscript. We agree with the Reviewer’s observation. Accordingly, we modified the “older” references in the “Introduction” section. Particularly: · References from [30] to [44] were removed because Table 1 was moved to the “Supplementary Material”; · References [45] and [46] were removed and not replaced; · References [4] – [5] – [6] – [7] – [9] were replaced with more recent references.
|
|
Authors have described more about TKI, so it will be great to expand the effects of ICI in bone metabolism used for renal cell carcinoma. |
We thank the Reviewer for this observation. Unfortunately, few studies are available concerning this topic. However, we expanded the section by adding 3 references. |
|
In the discussion part, it is suggested to add and explain, what could be an alternate to overcome hypophosphatemia, hypocalcemia, and hypercalcemia in RCC patients after treating with TKI and ICI. |
We thank the Reviewer for this observation. In the Discussion section, we added some suggestions for the treatment of altered calcium-phosphorus balance associated with anticancer therapies. |
|
The language expression of the manuscript needs to be further simplified and polished. |
We thank the Reviewer for this observation. The manuscript has been revised to improve the English language. |
Reviewer 2 Report
Dear editors and authors:
The authors reviewed the impact of tyrosine kinase inhibitors (TKI) and immune-checkpoint inhibitors (ICI)on bone and bone metabolism in patients with advanced renal cell carcinoma. However, there was a little information about the side effect of TKI and ICI in the advanced RCC patients and a few important questions need to be clarified. Based on the criteria of journal cancers, I suggested reconsider after major revision.
Major comments:
1. There was a little information about the side effect of TKI and ICI in the advanced RCC patients. In the introduction part, the authors should add information of the most common dermatologic side effects during the TKI EGFR treatment, pathogenesis, incidence and management of the side effect. Because, TKI inhibition in bone metabolism is also a side effect of the drug.
2. As we know, the TKI inhibition in bone metabolism is also a side effect of the drug. The authors only focused on advanced renal cell carcinoma patients in the review, is there any special characteristics of TKI and ICI treatment in the advanced RCC patients? Is the incidence of the side effect (inhibition bone metabolism) in advanced RCC patients higher than other cancer patients? Or is the incidence of side effect higher in advanced RCC patients with TKI and ICI therapy? Please clarify.
3. What is the management for the side effect of TKI and ICI? please discuss this in the discussion.
4. There were a few references in recent 5 years, please update.
Author Response
|
Reviewer #2
The authors reviewed the impact of tyrosine kinase inhibitors (TKI) and immune-checkpoint inhibitors (ICI)on bone and bone metabolism in patients with advanced renal cell carcinoma. However, there was a little information about the side effect of TKI and ICI in the advanced RCC patients and a few important questions need to be clarified. Based on the criteria of journal cancers, I suggested reconsider after major revision.
|
|
|
Observations/suggestions |
Authors’ answers |
|
There was a little information about the side effect of TKI and ICI in the advanced RCC patients. In the introduction part, the authors should add information of the most common dermatologic side effects during the TKI EGFR treatment, pathogenesis, incidence and management of the side effect. Because, TKI inhibition in bone metabolism is also a side effect of the drug. |
We thank the Reviewer for his/her general comments concerning our manuscript.
We agree with the Reviewer’s observation. Accordingly, we reported in the “Introduction” section a summary of the most common adverse events related to TKI (including dermatologic side effects) and immune checkpoint inhibitors as well. |
|
As we know, the TKI inhibition in bone metabolism is also a side effect of the drug. The authors only focused on advanced renal cell carcinoma patients in the review, is there any special characteristics of TKI and ICI treatment in the advanced RCC patients? Is the incidence of the side effect (inhibition bone metabolism) in advanced RCC patients higher than other cancer patients? Or is the incidence of side effect higher in advanced RCC patients with TKI and ICI therapy? Please clarify. |
We thank the Reviewer for this observation. We specify in the Discussion section that RCC patients on ICI or TKI treatment could represent a particular model to evaluate the effects of these classes of drugs on bone metabolism. In fact, chronic kidney disease represents per se a risk factor for altered bone metabolism-associated diseases. No specific studies are available concerning the incidence of bone adverse effects in RCC patients on anticancer therapies and focused studies are necessary. |
|
What is the management for the side effect of TKI and ICI? please discuss this in the discussion. |
We thank the Reviewer for this observation. We added a new “Discussion” paragraph, where we discussed the overall management of the side effects of TKI and ICI, also providing our perspectives on the effects of these drugs on bone metabolism. |
|
There were a few references in recent 5 years, please update. |
We thank the Reviewer for this observation. Accordingly, we have updated some of the references in the manuscript. |
Reviewer 3 Report
Thank you for your submission. Please find my comments below.
1. Table 1 does not add much to the manuscript because most of the data are not reported. This can be deleted or moved to a supplementary figure.
2. We need a comprehensive figure on the action of the various drugs ( TKI versus VGEF versus Immunotherapy) on how they are affecting bone metabolism.
3. What do the authors suggest on the recommendations to prevent osteoporosis while RCC patients are getting treated, any recommendations on DEXA scan/VitamD supp. Please, share your opinion with evidence (This may be helpful to refer-https://www.nature.com/articles/s41585-018-0034-9).
4. Please highlight if you discussed the effect of everolimus and bone metabolism,? is there any data on RCC
Author Response
|
Reviewer #3
Thank you for your submission. Please find my comments below.
|
|
|
Observations/suggestions |
Authors’ answers |
|
Table 1 does not add much to the manuscript because most of the data are not reported. This can be deleted or moved to a supplementary figure. |
We agree with the Reviewer’s observation. Accordingly, we moved Table 1 in the Supplementary Material. |
|
We need a comprehensive figure on the action of the various drugs ( TKI versus VGEF versus Immunotherapy) on how they are affecting bone metabolism. |
We thank the Reviewer for this observation. A figure has been added. |
|
What do the authors suggest on the recommendations to prevent osteoporosis while RCC patients are getting treated, any recommendations on DEXA scan/VitamD supp. Please, share your opinion with evidence (This may be helpful to refer-https://www.nature.com/articles/s41585-018-0034-9). |
We thank the Reviewer for this observation. In the Discussion section, we added generic suggestions about the treatment of osteoporosis in RCC patients. We thank the Reviewer for the suggested article. We added this reference. |
|
Please highlight if you discussed the effect of everolimus and bone metabolism,? is there any data on RCC |
We thank the Reviewer for this observation. We highlighted the role of mTOR inhibitors in the “Introduction” paragraph, discussing their effects on bone metabolism as well. We added a specific comment also in the Discussion section. |
Round 2
Reviewer 2 Report
Dear editors :
The authors improved the manuscript significantly, I suggested accepting the manuscript.
Reviewer 3 Report
Thank you for the revision